# Antioxidant, Antimicrobial and Antibiofilm Activity of Coriander (*Coriandrum sativum* L.) Essential Oil for Its Application in Foods

**DOI:** 10.3390/foods9030282

**Published:** 2020-03-04

**Authors:** Miroslava Kačániová, Lucia Galovičová, Eva Ivanišová, Nenad L. Vukovic, Jana Štefániková, Veronika Valková, Petra Borotová, Jana Žiarovská, Margarita Terentjeva, Soňa Felšöciová, Eva Tvrdá

**Affiliations:** 1Department of Fruit Sciences, Viticulture and Enology, Faculty of Horticulture and Landscape Engineering, Slovak University of Agriculture, Tr. A. Hlinku 2, 94976 Nitra, Slovakia; l.galovicova95@gmail.com; 2Department of Bioenergy, Food Technology and Microbiology, Institute of Food Technology and Nutrition, University of Rzeszow, Zelwerowicza St. 4, 35601 Rzeszow, Poland; 3Department of Technology and Quality of Plant Products, Faculty of Biotechnology and Food Sciences, Slovak University of Agriculture, Tr. A. Hlinku 2, 94976 Nitra, Slovakia; eva.ivanisova@uniag.sk; 4Department of Chemistry, Faculty of Science, University of Kragujevac, 34000 Kragujevac, Serbia; nvchem@yahoo.com; 5AgroBioTech Research Centre, Slovak University of Agriculture, Tr. A. Hlinku 2, 94976 Nitra, Slovakia; jana.stefanikova@uniag.sk (J.Š.); veronika.valkova@uniag.sk (V.V.); petra.borotova@uniag.sk (P.B.); 6Department of Plant Genetics and Breeding, Faculty of Agrobiology and Food Resources, Slovak University of Agriculture, Tr. A. Hlinku 2, 94976 Nitra, Slovakia; jana.ziarovska@uniag.sk; 7Institute of Food and Environmental Hygiene, Faculty of Veterinary Medicine, Latvia University of Life Sciences and Technologies, K. Helmaņa iela 8, 3004 Jelgava, Latvia; margarita.terentjeva@llu.lv; 8Department of Microbiology, Faculty of Biotechnology and Food Sciences, Slovak University of Agriculture, Tr. A. Hlinku 2, 94976 Nitra, Slovakia; sona.felsociova@uniag.sk; 9Department of Animal Physiology, Faculty of Biotechnology and Food Sciences, Slovak University of Agriculture, Tr. A. Hlinku 2, 94976 Nitra, Slovakia; eva.tvrda@uniag.sk

**Keywords:** coriander, *Bacillus*, *Penicillium*, *Stenotrophomonas*, antioxidant activity, biofilm formation, mass spectrometry, wooden and glass surfaces

## Abstract

The aim of this study was to assess the chemical composition, antioxidant, antimicrobial and antibiofilm activity of the *Coriandrum sativum* essential oil. Changes in the biofilm profile of *Stenotropomonas maltophilia* and *Bacillus subtilis* were studied using MALDI-TOF MS Biotyper on glass and wooden surfaces. The molecular differences of biofilms in different days were observed as well. The major volatile compounds of the coriander essential oil in the present study were *β*-linalool 66.07%. Coriander essential oil radical scavenging activity was 51.05% of inhibition. Coriander essential oil expressed the strongest antibacterial activity against *B. subtilis* followed by *S. maltophilia* and *Penicillium expansum.* The strongest antibiofilm activity of the coriander essential oil was found against S. *maltophilia.* A clearly differentiated branch was obtained for early growth variants of *S. maltophilia* in case of planktonic cells and all experimental groups and time span can be reported for the grouping pattern of *B. subtilis* preferentially when comparing to the media matrix, but without clear differences among variants. The results indicate that coriander was effective against the tested *Penicillium expansum* in the vapor phase after 14 days with MID_50_ 367.19 and MID_90_ 445.92 µL/L of air.

## 1. Introduction

Coriander (*Coriandrum sativum* L.) belongs to the *Apiaceae* family and it is an extensively used medicinal plant with nutritional and medicinal properties. The extracts and essential oils of coriander have exhibited antibacterial, antioxidant, free radical, antidiabetic, anticancer and antimutagenic activities. The nature of the solvent was found to be the principal factor in the extraction of antioxidants and bioactive compounds from coriander. All parts of coriander are edible, but their flavours and applications are different. Reported studies on the antioxidant activities of *C. sativum* were mostly focused on the aerial parts [1].

Biofilm formation is a crucial topic for food production and medicine. Understanding of the formation of biofilms is important in the research focused on effective strategies to combat harmful biofilms. Strict requirements have been elaborated to control biofilm formation by pathogenic bacteria, and were based on clinically relevant bacteria [2,3,4]. Previous studies on biofilm development have involved mostly *Staphylococcus epidermidis*, *Staphylococcus aureus*, and *Pseudomonas aeruginosa* species interaction. *Stenotrophomonas maltophilia* is an important nosocomial pathogen, the pathogenesis of which is based on an establishment of the biofilms on mucosal surfaces or prostheses. Clinical and environmental *S. maltophilia* isolates have been reported to adhere to abiotic [5,6,7] and biotic [8] surfaces with little information available about the biofilm formation and antibiotic activity against *S. maltophilia* biofilms.

*Bacillus subtilis* is an important bacterium for food processing industry because of its ability to form rough biofilms at the air-liquid interface rather than on solid-liquid phase due to the aerotaxis. Biofilms produced by *B. subtilis* and related species have shown to have a direct impact on the control of plant pathogens and the steel corrosion. The prevention of formation of harmful biofilms and their replacement with beneficial biofilms formed by industrial bacteria may lead to new knowledge in the field of biotechnology [9].

Essential oils were proved to possess a strong antimicrobial activity that is used in the food industry to control the microbial spoilage, food safety by inhibition of foodborne pathogens, and the prolongation of the product shelf life [10]. Coriander essential oil (CEO) has been capable to prevent biofilm formation of *Acinetobacter baumannii*, and hence could be considered as a potential alternative to antimicrobials. CEO was shown to promote the disintegration of *A. baumannii* biofilm which is important for the prevention of the pathogen dissemination [11].

The morphology and structure of bacterial biofilms have been characterized by microscopic techniques previously [12,13,14]. Recently, the matrix-assisted laser desorption ionization time-of-flight mass spectrometry (MALDI-TOF MS) has been applied for the description of the molecular profile of biofilm-forming bacteria [15,16]. Furthermore, the discrimination of different stages within the bacterial biofilm formation based on MALDI-TOF MS profiling was reported [17].

The aims of the present study were (i) to examine the antioxidant, antimicrobial and antibiofilm activity of the coriander essential oil against *Stenotropomonas maltophilia* and *Bacillus subtilis*, (ii) to evaluate the molecular profiles of *S. maltophilia* and *B. subtilis* biofilms on glass and wood surfaces using MALDI-TOF MS Biotyper and (iii) to detect the efficacy of the coriander essential oil in the control of *Penicillium expansum* growth by contact vapor method.

## 2. Materials and Methods

### 2.1. Origin of Microorganisms

The biofilm-forming strains of *Stenotropomonas maltophilia* and *Bacillus subtilis* were obtained from the milk industry. The strains were tested for antibiotic resistance, antimicrobial, antioxidant and biofilm activity. The fungus *Penicillium expansum* MK-SF 33 was isolated and identified from grapes.

### 2.2. Essential Oil

Coriander essential oil (CEO) was purchased from Hanus, a.s. (Slovakia) and subsequently tested chromatographically. CEO was prepared by water vapor distillation of the dried fruit. The sample was stored in the dark at 4 °C.

### 2.3. Chemical Composition of the Essential Oil

The gas chromatography coupled with mass spectrometry (GC-MS, Agilent 7890B, Agilent 5977A, Agilent Technologies Inc., Palo Alto, CA, USA) and CombiPal autosampler 120 (CTC Analytics AG, Zwingen, Switzerland) was used for semi-quantitative composition of the coriander essential oil with modifications [18]. The column DB-WAXms^®^ with polyethylene glycol stationary phase (30 m × 0.25 mm × 0.25 µm) (Agilent Technologies, Santa Clara, CA, USA) was used. The identification of compounds was carried out by comparing the obtained mass spectra (over 80% match) with the NIST^®^ 2017 commercial database (National Institute of Standards and Technology, Gaithersburg, MD, USA). The semi-quantitative content of the determined compounds was calculated by dividing the individual peak area (excluded by solvent peak area) by the total area of all peaks. The experiment was performed in duplicate.

### 2.4. Radical Scavenging Activity—DPPH Method

The radical scavenging activity of the essential oil was measured using 2,2-diphenyl-1-picrylhydrazyl (DPPH) method according to Sanchéz-Moreno et al. [19] with slight modifications. The sample (0.1 mL) was mixed with 3.9 mL of DPPH solution (0.025 g DPPH in 100 mL methanol). Absorbance of the reaction mixture was determined with the Jenway spectrophotometer (6405 UV/Vis, England) at 515 nm. Trolox (10–100 mg/L; R2 = 0.998) was used as standard, and the results were expressed in mg/L of Trolox equivalents. The scavenging activity was calculated as percentage (AA%) and determined according to the following formula:AA% = [(A0 − AAT)/A0 × 100]
where A0 is the absorbance of the control reaction (DPPH radical); A1 is the absorbance of the tested sample.

The experiment was performed in triplicate.

### 2.5. Antimicrobial Activity with Disc Diffusion Method

The antibacterial activity was examined by the agar disc diffusion method. Bacterial strains of *S. maltophilia* and *B. subtilis* were incubated in the Mueller Hinton broth (MHB, Oxoid, Basingstoke, UK) at 37 °C for 24 h. *P. expansum* were incubated in the Sabouraud broth (SB, Oxoid, Basingstoke, UK) at 25 °C for 48 h. *S. maltophilia, B. subtilis* and *P. expansum* isolates were identified by MALDI-TOF MS Biotyper (Brucker, Daltonic, Germany) with high scores, 2.345, 2.234 and 2.425, respectively. Microbial suspensions in saline water of 0.5 McFarland turbidity standards (densitometer, Erba Lachema s.r.o., Brno, Czech Republic) were streaked onto Mueller-Hinton agar (MHA) and Sabouraud agar with a sterile swab. A sterile filter disc (diameter 6 mm, Whatman paper N. 1) was impregnated with EOs (10 µL/disc). The inoculated MHAs were placed at 4 °C for 2 h and then incubated at 37 °C, resp. 25 °C for 18–24 h, resp. 48 h. The antimicrobial activity was evaluated by measuring the zones of growth inhibition. Gentamicin (10 µg, Oxoid, Basingstoke, UK) was used as the positive control. The experiment was performed in triplicate.

### 2.6. Minimum Biofilm Inhibitory Concentration (MBIC)

The minimum inhibitory biofilm concentration (MBIC) was performed using the agar microdilution method [20]. The bacterial culture was cultured in the Muller Hinton broth (MHB) at 37 °C for 24 h. One hundred μL of the bacterial suspension with a density of 10^8^ CFU/mL were inoculated into each cell of a 96-well microtitration plate. Subsequently, 100 µL of the essential oil with a concentration from 0.3125 μL to 10 μL per well was added. As a negative control, a mixture of MHB with essential oil was selected, while a mix of MHB with bacterial suspension was used as a control of maximum growth. Following cultivation at 37 °C for 24 h, the supernatant from the plate wells was poured away and the wells were washed three times with 250 µL of saline solution. After washing, the plates were dried for 30 min at room temperature and stained with crystal violet (200 µL of 0.1% (*w/v*)) for 15 min. The plates were washed repeatedly with distilled water and allowed to dry. The samples were resolubilized with 200 μL of 33% acetic acid [21]. The absorbance was measured with the Glomax plate spectrophotometer (Promega Inc., Madison, WI, USA) at 570 nm. The concentration of essential oil at which the absorbance was less than or equal to the negative control was determined as MBIC. The test was performed in triplicate and the average (n = 3) was used for further calculations.

### 2.7. Analysis of the Biofilm Development Phases and Evaluation of Molecular Differences on Different Surfaces Using MALDI-TOF MS Biotyper

Different stages of the biofilm development during growth and the application of MALDI-TOF MS Biotyper for the evaluation of molecular differences in biofilms grown on different surfaces were evaluated. Growing planktonic cells were used for biofilms. The experiment was performed in 50 mL polypropylene tubes. Each tube contained 20 mL MHB, a glass slide and a wooden toothpick. Prior to the inoculation, cells were incubated in MHB at 37 °C for 24 h. Ten μL of suspension was added into the polypropylene tubes. For the evaluation of the effect of coriander essential oil, the MHB was enriched with 0.1% of essential oil. The prepared samples were incubated at 37 °C in a shaker with a 45° inclination at 170 rpm. The biofilm and planktonic cell samples were collected on days 3, 5, 7, 9, 12, and 14 of cultivation. Biofilm samples from the microscope slide and toothpick were taken with a sterile cotton swab and transferred directly on the MALDI-TOF MS Biotyper plate. Planktonic cells were collected by centrifugation of 300 µL of the medium previously collected to an Eppendorf tube at 3000 g for 3 min. The supernatant was resuspended in 25 µL of ultrapure water and the washing procedure was performed three times. Next, 1 μL of the suspension was applied to the MALDI-TOF MS Biotyper plate (Bruker Daltonics, Bremen, Germany) in duplicate.

The samples were covered with 1 µL of an acyano-4-hydroxy-cinnamic acid matrix (10 mg/mL) and dried at room temperature. After crystallization, the samples were processed with MALDI-TOF MicroFlex (Bruker Daltonics) linear and positive mode for the range of *m/z* 200–2000. The spectra were obtained by an automatic analysis and the same sample similarities were used to generate the standard global spectrum (MSP), 40 spectra of MALDI Biotyper 3.0 software were used for the demonstration of all the phases of biofilm development (Bruker Daltonics). MSP spectra were processed by a standard procedure using the MALDI Biotyper dendrogram method and the most representative sample of each spectrum was selected based on common characteristics. The spectra obtained in the present study were compared with the FlexAnalysis 3.0 database (Bruker Daltonics). From the spectra generated by MALDI Biotyper 3.0, 11 MSP were grouped into dendrograms using Euclidean distances [17].

### 2.8. Bread Making Process

The baking formula consisted of wheat flour T650 (250 g), water (150 mL), saccharose (2.5 g), salt (5 g) and yeast (5 g). All the ingredients were mixed in a spiral mixer (Diosna SP 12 D, Diosna, Osnabrück, Germany) for 6 min. The dough was placed into an aluminum vessel and transferred into a fermentation cabinet (MIWE cube, Pekass s.r.o., Plzeň, Czech Republic) at 32 °C (85 % relative humidity) for 40 min. The loaves were baked at 180 °C for 17 min with the addition of 160 mL water and at 210 °C for 10 min in a steamy oven (MIWE cube, Pekass s.r.o., Plzeň, Czech Republic). After baking, the bread was left to stand at laboratory temperature for 2 h.

### 2.9. Water Activity and Moisture Content

Water activity (a_w_) of breadcrumbs was determined with the Lab Master a_w_ Standard (Novasina, Lachen, Switzerland) by placing 2.0 g of sample in a sample pan. The a_w_ was measured automatically at 25 °C.

The moisture content of the cooled crumbs was determined using the Kern DBS 60-3 moisture analyzer (Kern & Sohn GmbH, Balingen, Germany) by placing 1.0 g of the sample onto the sample plate, measurement was done at 120 °C.

### 2.10. In-Situ Antifungal Analysis on Bread

The bread was cut into slices (height of 150 mm) and placed into 0.5 L sterile glass jars (Bromioli Rocco, Italy). Fungal spore suspension of each strain (final concentration of spores 1 × 10^6^ spores/mL) was diluted in 20 mL of sterile phosphate-buffered saline with 0.5% Tween 80 by adjusting the density to 1–1.2 McFarland. Then, 5 µL of the inoculum was added on top of the bread at three different places. Next, 100 µL of the solution (coriander essential oil of 125, 250 and 500 µL/L concentration + ethyl acetate) was evenly distributed on a sterile paper-filter disc (6 cm) and inserted into the cover of the jar. The treatment with essential oil was not done for the control group. Jars were hermetically closed and kept at 25 °C ± 1 °C for 14 days in the dark. After incubation, the colonies with visible mycelial growth and visible sporulation were counted.

### 2.11. Statistical Analysis

All measurements and analyses were carried out in triplicate. The experimental data were evaluated by basic statistical variability indicators using the Microsoft™ Excel^®^ program. The results of the MBIC50 and MBIC90 value (concentration causing 50% and 90% reduction of bacterial biofilm growth) were estimated by logit analysis.

## 3. Results

### 3.1. Chemical Composition of Coriander Essential Oil

Semi-quantitative composition of the coriander essential oil was characterized by gas chromatography coupled with mass spectrometry (GC-MS), first. The major volatile compounds of the analyzed coriander essential oil based on their decreased percentages were *β*-linalool 66.07%, camphor 8.34%, geranyl acetate 6.91% and cymene 6.35% (Table 1).

### 3.2. Antioxidant, Antimicrobial and Antibiofilm Activity of Coriander Essential Oil

The basic characteristics relevant to bio-activity was studied in the next step of this complex evaluation of coriander essential oil. In our study, the coriander essential oil radical scavenging activity was 39.38 mg TEAC/L (Trolox equivalent antioxidant activity) equivalent to 51.05% of inhibition.

In the next step, antibacterial and antibiofilm activity was evaluated for the most relevant bacteria species. Coriander oil expressed the strongest antibacterial activity against *B. subtilis* (10.69 ± 0.47) followed by *S. maltophilia* (9.22 ± 0.08 mm) and *P. expansum* (8.99 ± 0.08 mm), the antibacterial activity of gentamicin was 26.33 ± 1.53 mm. The MBIC_50_ and MBIC_90_ of S. *maltophilia* of the coriander essential oil were 7.49 μL/mL and 7.96 μL/mL, respectively. The MBIC_50_ and MBIC_90_ of *B. subtilis* were 7.42 μL/mL and 6.95 μL/mL, respectively.

### 3.3. Analysis of Biofilm Development Stages and Evaluation of Molecular Differences on Different Surfaces Using MALDI-TOF MS Biotyper

The spectra of *S. maltophilia* during the biofilm development stages in the experiment are shown in Figure 1. The spectra were aligned together in pairs according to their growth stage on different surfaces with the exception of spectra of planktonic cells which were obtained from the culture medium.

The spectra of *B. subtilis* during the biofilm development stages in the experiment is shown Figure 2. The spectra were aligned in pairs according to the stage of growth on different surfaces with the exception of spectra of planktonic cells which were obtained from the culture medium.

Mass spectra of *S. maltophilia* on day 3 of the experiment for the planktonic cells, control and experimental groups were compared (Figure 1). The mass spectra of *S. maltophilia* on day 5 of the experiment revealed differences between planktonic cells, control and experimental groups (Figure 1) while the spectra of the experimental group failed to correspond to planktonic cells spectra but the control group was comparable to the planktonic cells. The same trend was observed on days 7, 9, 12 and 14 of the experiment with a decrease in the mass spectra for the experimental group as compared to the planktonic cells and the control group (Figure 1).

Our results indicate that the MSPs of the groups were distinguishable by MALDI profiling, since they could be separated into different clusters (Figure 3, Figure 4, Figure 5 and Figure 6). It can be observed that the planktonic stage of *S*. *maltophilia* showed the highest difference between the 3 and 5 days old biofilms in the MSP distance level. Moreover, the clustering of the planktonic, 9-day and 12-day groups in the same main cluster indicate the decreased divergence as the biofilm grows older (Figure 3).

MSP grouping of *S. maltophilia* for planktonic cells and all experimental groups showed that the young P, SMD 3 and SMS 3 biofilms were the most distant group, while the biofilm formation profile of *S. maltophilia* was balanced in the grouping manner (Figure 4).

The biofilm age was important for the grouping of *B. subtilis* (Figure 5 and Figure 6).

### 3.4. Effect of Coriander Essential Oil in the Selected Characteristics of Bread and Antifungal Effect of CEO

The potential of practical application of coriander oil was evaluated in the 14 days stored bread finally. Fresh bread was baked following the very basic formula. The moisture content that was obtained for this bread was 41.467 ± 0.881% and a_w_ comprised 0.945 ± 0.002. The results of performed in-situ antifungal analysis on bread indicate good antifungal activity, as the MID_50_ and MID_90_ of the coriander essential oil for *Penicillium expansum* on the bread after 14 days were 367.19 and 445.92 µL/L of air, respectively.

## 4. Discussion

The essential oils and extracts of aromatic plants and spices contain biologically active components with antibacterial, antifungal and antioxidant activities that have allowed to use them widely for pharmaceutical and food processing industry needs. Coriander (*C. sativum* L.) of the family *Umbelliferae*/*Apiaceae* is a glabrous, aromatic, herbaceous annual plant with a long history of application in culinary and serving as a source of aroma compounds and essential oils. *C. sativum* may be applied for food preparation as a flavoring agent and adjuvant, and for the prevention of food borne diseases and food spoilage because of the antibacterial properties of its essential oil [22].

The major volatile compounds of the coriander essential oil in the present study were *β*-linalool 66.07%, camphor 8.34%, geranyl acetate 6.91% and cymene 6.35%. Shahwar et al. [23] reported linalool, *γ*-terpinene, *α*-pinene, camphor, decanal geranyl acetate, limonene, geraniol, camphene and d-limonene to be the main volatile compounds for seeds, while (E)-2-decenal, linalool, (E)-2-dodecenal, (E)-2-tetradecenal, 2-decen-1-ol, (E)-2-undecenal, dodecanal, (E)-2-tridecenal, (E)-2-hexadecenal, pentadecenal and α-pinene were the main compounds of the coriander essential oils obtained from leaves. The growing seasons, growth stage and climatic conditions have been proved to influence the content and chemical composition of the essential oil [24,25,26,27]. The major volatile compounds of the coriander essential oil in our study were *β*-linalool 66.07%, camphor 8.34%, geranyl acetate 6.91% and cymene 6.35%.

According to Shahwar et al. [23], the activity of the coriander essential oil determined by DPPH was 66.48%. In our study, the coriander essential oil radical scavenging activity was 51.05% of inhibition. Singh et al. [28] found the radical scavenging activity as IC50 in the amount of 47.2 μg/mL using the same method. Coriander, like many other spices, contains natural antioxidants, which can delay or prevent the spoilage of seasoned food, which is why the natural antioxidants are widely applied in food industry. Darughe et al. [29] observed the effect of coriander essential oil as a natural antioxidant in cakes and proposed it as potential substitutes of synthetic antioxidants in the food preservation. As reported by Marangoni and Moura [30] salami with the addition of the coriander essential oil led to a reduction in lipid oxidation and thus increased the shelf-life.

In a previous study, the effect of the coriander essential oil was compared with essential oils prepared from other plants against 11 different bacterial and three fungal species reported to be involved in food poisoning and decay. The coriander essential oil exhibited the strongest antibacterial activity against all while thyme and spearmint oils inhibited fungal species [31]. The most effective antifungal activity among the family *Apiaceae* has been previously reported for coriander essential oils against *Aspergillus flavus* and aflatoxin production at all concentrations studied [32]. The authors of that particular study recommended that an amount of 1000 ppm of the coriander essential oil could be suitable as a food additive for the protection against biodeterioration caused by fungi as well as against aflatoxin contamination. In our study, coriander oil expressed the strongest antibacterial activity against *B. subtilis* followed by *S. maltophilia* and *P. expansum.*

Duarte et al. [33] identified the antibacterial potential of the coriander essential oil against the biofilm produced by *Acinetobacter baumannii* at 4 μL/mL for MIC and MBC. In our study we determined the antimicrobial activity with the disc diffusion method. In the study of Bazargani and Rohloff [34], antiadhesion tests using the crystal violet assay for the evaluation of the inhibition of cell attachments were carried out for the coriander essential oil. They found different effects of the coriander essential oil on the growth and development of the biofilm with at least 50% reduction in the cell attachment with a full inhibition of *S. aureus*. According to Bazargani and Rohloff [34], the coriander essential oil induced the inhibition of biofilm formation against *S. aureus* up to 91%. MIC value in this study ranged from 0.8 to 0.63 μL/mL of two bacterial strains with different essential oils. In our study MBIC_50_ values were bigger from 7.42 to 7.49 µL/mL.

The MBIC_50_ and MBIC_90_ of S. *maltophilia* of the coriander essential oil were evaluated for the first time.

In our study the antibiofilm activity of coriander essential oil with MALDI TOF MS Biotyper was evaluated. Pereira et al. [17] reported that MALDI-TOF MS Biotyper was a useful tool for the identification of microorganisms. Di Bonaventura et al. [35] showed that the MALFI-TOF MS is a suitable tool to detect different stages of the biofilm development and to assess some of the characteristics of surfaces on which the bacteria were grown: bacteria on the plastic surface maturated faster until achieving the dispersed stage in a shorter time. *S. maltophilia* was found to attach to polystyrene within 2 h of incubation followed by biofilm formation, to produce the maximum growth at 24 h [35]. Twenty-four pulsotypes of *S. maltophilia* isolates with a different capacity to the biofilm production were reported previously [36]. In this study *S. maltophilia* strain produced biofilms similarly to our experiments.

*B. subtilis* is a well-known model microorganism to study the molecular principles of biofilm development. *B. subtilis* growth kinetics and morphological features were characterized with a colony type biofilm formation [37]. Biofilm of *B. subtillis* is quickly developing into a three-dimensional complex structure with an unchanged core and expansion, which is mostly related to the outer cell subpopulations enlargement. In our study the biofilm formation growth from 0 to 12 days. Duarte et al. [33] found at least 85% inhibition of the biofilm formation in experiments with coriander essential oils and their effect on *Acinetobacter baumannii’s* plankton and biofilm cells.

Bread is a food product with intermediate moisture that is prone to molding spoilage and is widely used in the human diet. The main species of fungi which may cause bread spoilage include *Penicillium, Aspergillus, Monilia, Mucor, Endomyces, Cladosporium, Fusarium* and *Rhizopus*. In our study, *Penicillium expansum* was chosen for the experimental design due to the fact that it is more resistant than other representative *Penicillium* species found in breads [38,39,40,41]. Wheat bread with a 41.467 ± 0.881% moisture content was used in the current study as bread goods with a moisture content from 35% to 42% are prone to microbial spoilage within a few days or weeks [42]. According to Cauvain et al. [43], the moisture content and a_w_ are determinants of the shelf-life of bakery products. Water activity represents the ratio of the vapor pressure of water over a substrate compared to that over pure water at the same temperature and pressure. Microscopic fungi grow in products with low levels of a_w_ ranging between 0.80 and 0.65 [44]. The bread with high a_w_ (0.945 ± 0.002) was used in agreement with Day et al. [42] (above 0.95).

Previously, the antifungal effect of the coriander essential oil for *Candida* spp. was reported [45]. Coriander exhibited an excellent antifungal activity against seed-borne pathogens of paddy depending on its concentration [46]. Zare-Shehneh et al. [47] recorded that the coriander leaf extract had fungicidal activity against *Penicillium lilacinum* and *Asperjilus niger* with MICs of 67.8 and 62.1 μg/mL, respectively.

## 5. Conclusions

Our results indicate that the coriander essential oil has the inhibiting effect on the biofilm formation. These findings were confirmed by the mass spectra analysis done by MALDI-TOF MS supporting the antibiofilm activity of the coriander essential oil. As such, MALDI-TOF MS profiling could be a useful technology in the analysis of biofilm profiles.

MALDI-TOF MS Biotyper application allows to distinguish between different phases of the biofilm with a possible application in food industry.

This study also demonstrates the antifungal effect of the coriander essential oil which can be used as an alternative to the chemical inhibition of the *Penicillium expansum* growth in bread.

## Figures and Tables

**Figure 1 foods-09-00282-f001:**
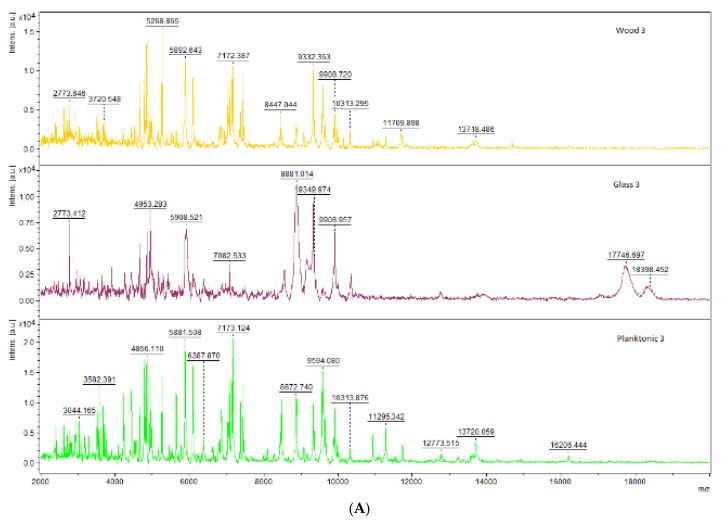
Representative MALDI-TOF mass spectra of *S. maltophilia*: (**A**) 3 days; (**B**) 5 days; (**C**) 7 days; (**D**) 9 days; (**E**) 12 days; and (**F**) 14 days.

**Figure 2 foods-09-00282-f002:**
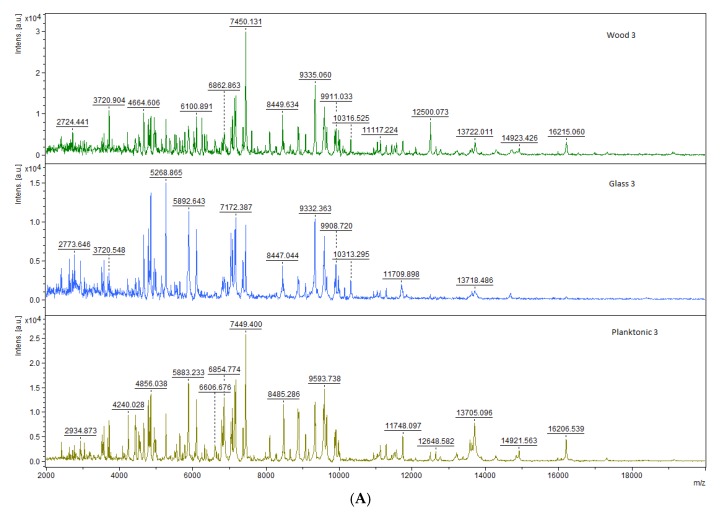
Representative MALDI-TOF mass spectra of *B. subtilis*: (**A**) 3 day; (**B**) 5 day; (**C**) 7 day; (**D**) 9 day; (**E**) 12 day; (**F**) 14 day.

**Figure 3 foods-09-00282-f003:**
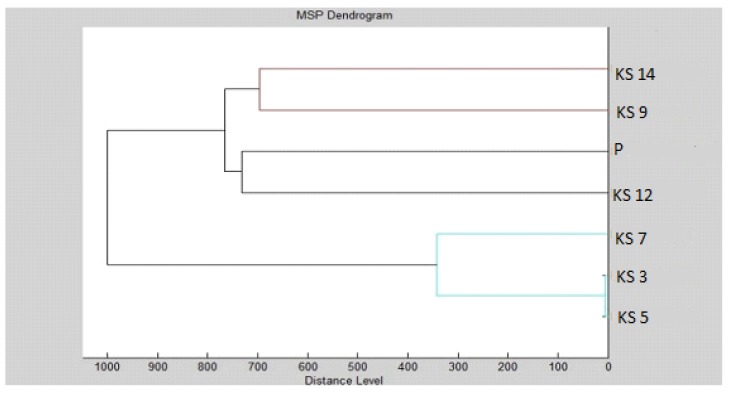
Dendrogram of *S*. *maltophilia* generated using the standard global spectrums (MSPs) for planktonic cells and control: K-control; S-*Stenotrophomonas maltophilia*; P-planktonic cells.

**Figure 4 foods-09-00282-f004:**
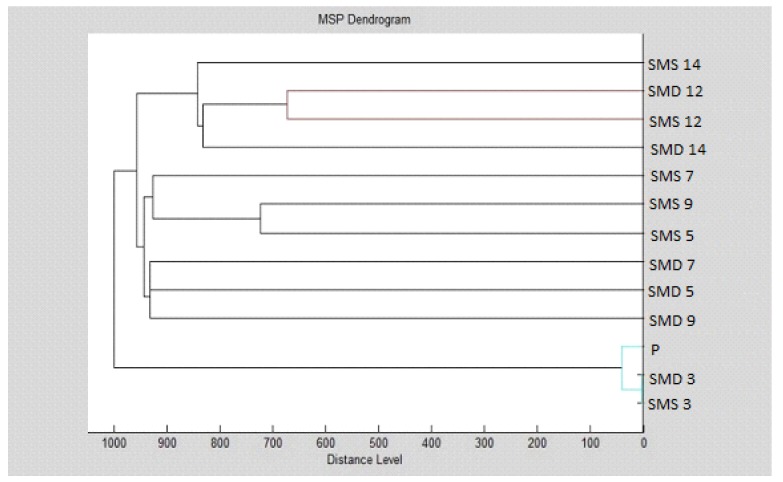
Dendrogram of *S*. *maltophilia* with MSP for planktonic cells and all experimental groups: SM-*Stenotrophomonas maltophilia*; S-glass; D-wood; P-planktonic cells.

**Figure 5 foods-09-00282-f005:**
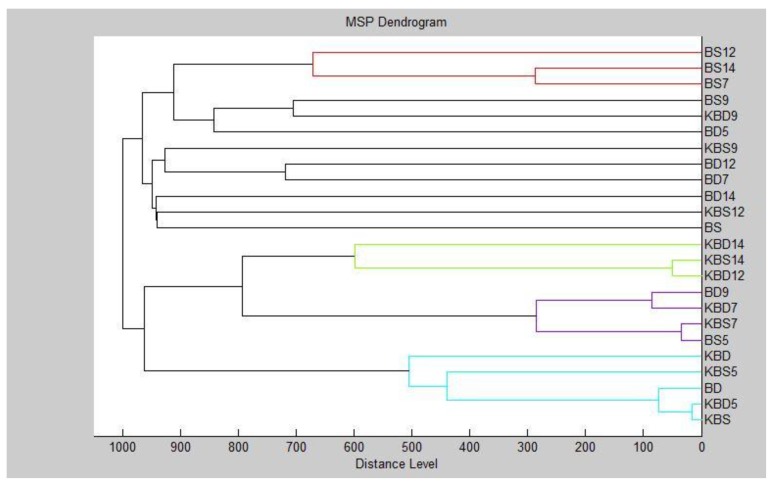
Dendrogram of *B*. *subtilis* generated using the MSPs for all experimental group: B-*B. subtilis*; K-control; S-glass; D-wood.

**Figure 6 foods-09-00282-f006:**
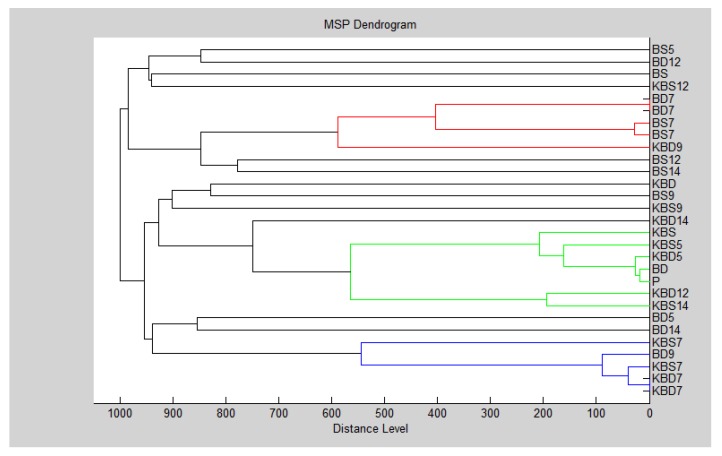
Dendrogram of *B*. *subtilis* generated using the MSPs for planktonic cells and control. Symbols in the abbreviations are as follows: B-*B. subtilis*; K-control; S-glass; D-wood; P-planktonic cells.

**Table 1 foods-09-00282-t001:** Relative chemical composition of coriander essential oil (CEO) obtained by GC-MS (gas chromatography coupled with mass spectrometry) analysis.

Name	Synonyms	TIC (Total Ion Chromatogram)% Area
2-myristynoyl pantetheine		0.35
*β*-myrcene		0.42
D-limonene		2.93
p-mentha-1,4-diene	*γ*-terpinene	1.96
cymene		6.35
1,2-oxolinalool		2.44
camphor	(+)-2-bornanone	8.34
(+/−)-linalool	*β*-linalool	66.07
*α*-terpineol		0.88
geranyl acetate		6.91
citronellol		0.39
trans-geraniol	guaniol	2.57
	lemonol	
	geraniol	
	geranyl alcohol	
	neryl alcohol	
terpendiol		0.37

Values represent means of duplicate determinations (maximum relative standard deviation ±5%). TIC%-total ion chromatograms (%).

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
