# Peer review of "Antioxidant, Antimicrobial and Antibiofilm Activity of Coriander (Coriandrum sativum L.) Essential Oil for Its Application in Foods"

_foods, 2020, doi:10.3390/foods9030282_

Round 1
Reviewer 1 Report
The manuscript entitled “Antioxidant, antimicrobial and antibiofilm activity of coriander (Coriandrum sativum L.) essential oil for its application in foods” by Miroslava et al., evaluated here the describes here the Antioxidant, antimicrobial and antibiofilm of essential oils from coriander to be used as food supplements.
Although authors state here an effective approach, there are few points’ needs to be addressed or included in the study.
General points:
Manuscript is very confusing and not easy to follow, so authors are requested to present it clearly while resubmitting the MS.
References should be carefully checked and corrected.
Specific points:
Abstract: Authors are requested to include some of its key finding more in details under abstract section.
Corriander oil purchased was from which part of the plant. Was it extracted from seed or the leaves? Information from the company is required…
The major component of the essential oil determined by author is B-Linalool, does authors have any plans of checking the activity of this compound individually and compare it with the mixture of these compounds purchased as essential oil.
“Sing et al. [28] found the radical scavenging” Whereas Singh et al reference is number 30. And also It should be Singh et al., not Sing et al., Authors are requested to check all the reference as one mistake would create completely disorder of the reference.
“Coriander oil expressed the strongest antibacterial activity against B. subtilis (10.69±0.47) followed by S. maltophilia (9.22±0.08) and P. expansum (8.99±0.08), the antibacterial activity of gentamicin was 26.33±1.53” What are these values? In mm? or in concentration?
What was the concentrations used for antibacterial assay. Please provide the details…including for Gentamycin? Under method section, Authors stated that 10ug of Gentamycin and 10uL of EO was used…It’s also suggested to use variable concentrations to know MIC values.
100ul of EO was used for MBIC testing….Why? if 10ul is already showing significant antibacterial activity, then the antibiofilm activity might be due to inhibitory activity rather than biofilm inhibitory activity. Hence, as stated before it is suggested to determine the non-inhibitory activity of EO by MIC and then use the Non-inhibitory concentration (NIC) to check for anti-biofilm activity.
“The MBIC50 and MBIC90 of S. maltophilia of the coriander essential oil were 7.49 μL/mL and 7.96 μL/mL, respectively. The MBIC50 and MBIC90 of B. subtilis were 7.42 μL/mL and 6.95 μL/mL, respectively.” How can author justify that it’s not an inhibitory activity?
There is direct switch between antibacterial to antifungal activity, which is not properly introduced/presented, which might be misleading/confusing for the reader and hence suggested to rewrite the introduction and explain clearly why only one fungi was tested for antifungal activity and how its related to bacterial as well as biofilm activity presented.
Again authors didn’t not present here the units “MID50 and MID90 of the coriander essential oil for Penicillium expansum on the bread after 14 days 365 were 367.19 and 445.92, respectively”
Author Response
Dear reviewers and editor
RE: “Antioxidant, antimicrobial and antibiofilm activity of coriander (Coriandrum sativum L.) essential oil for its application in foods"
Manuscript ID: Foods 721388
We would like to thank the reviewers for their valuable comments and recommendations. The manuscript has been corrected in line with the comments of reviewers. All corrections are highlighted. English was improved throughout the manuscript.
Reviewer 1:
General points: Manuscript is very confusing and not easy to follow, so authors are requested to present it clearly while resubmitting the MS.
The manuscript reports different types of analysis performed for the CEO. We have performed some changes to make the results easier to follow.
References should be carefully checked and corrected.
References was checked and corrected.
Specific points:
Abstract: Authors are requested to include some of its key finding more in details under abstract section.
Units were provided.
Coriander oil purchased was from which part of the plant. Was it extracted from seed or the leaves? Information from the company is required…
On line 98 was write that essential oil of coriander was produced by water vapour distillation of the dried fruit.
The major component of the essential oil determined by author is B-Linalool, does authors have any plans of checking the activity of this compound individually and compare it with the mixture of these compounds purchased as essential oil.
The authors planed in the future test β-linalool for the antimicrobial activity. This data we still don’t have so we are not adding to manuscript.
“Sing et al. [28] found the radical scavenging” Whereas Singh et al reference is number 30. And also It should be Singh et al., not Sing et al., Authors are requested to check all the reference as one mistake would create completely disorder of the reference.
It was corrected as references Singh et al. 29.
“Coriander oil expressed the strongest antibacterial activity against B. subtilis (10.69±0.47) followed by S. maltophilia (9.22±0.08) and P. expansum (8.99±0.08), the antibacterial activity of gentamicin was 26.33±1.53” What are these values? In mm? or in concentration?
The values were in mm. It was corrected in manuscript.
What was the concentrations used for antibacterial assay. Please provide the details…including for Gentamycin? Under method section, Authors stated that 10ug of Gentamycin and 10uL of EO was used…It’s also suggested to use variable concentrations to know MIC values.
In material and method is disc diffusion method and for antimicrobial activity were used only clear essential oil 10 µL on disc. It is not concentration. For gentamycin was used antibiotics disc with 10 µg as positive control, it is concentration for disc.
100ul of EO was used for MBIC testing….Why? if 10ul is already showing significant antibacterial activity, then the antibiofilm activity might be due to inhibitory activity rather than biofilm inhibitory activity. Hence, as stated before it is suggested to determine the non-inhibitory activity of EO by MIC and then use the Non-inhibitory concentration (NIC) to check for anti-biofilm activity.
100 µL of essential oils is not concentration. It is amount for plates. Concentration is in line 142-143 0.3125 μL to 10 μL.
“The MBIC50 and MBIC90 of S. maltophilia of the coriander essential oil were 7.49 μL/mL and 7.96 μL/mL, respectively. The MBIC50 and MBIC90 of B. subtilis were 7.42 μL/mL and 6.95 μL/mL, respectively.” How can author justify that it’s not an inhibitory activity?
The results of the MBIC50 and MBIC90 value (concentration causing 50% and 90% reduction of bacterial biofilm growth) were estimated by logit analysis. MBIC50 for S. maltophilia has inhibitory activity but for B. subtilis MIC50 has 7.42 μL/mL and for MBIC90 6.95 μL/mL. That is mean that essential oil has only antibiofilm activity of MBIC50 with value 7.42 μL/mL and MBIC90 only 6.95 μL/mL
There is direct switch between antibacterial to antifungal activity, which is not properly introduced/presented, which might be misleading/confusing for the reader and hence suggested to rewrite the introduction and explain clearly why only one fungi was tested for antifungal activity and how its related to bacterial as well as biofilm activity presented.
In the study was primary aim antibiofilm activity and P. expansum was tested in food model, because journal topic is food. Evaluation of antibiofilm activity of coriander essential oil were used with MALDI TOF MS Biotyper in first time. Second reviewer wrote that introduction section is balanced and needs no changes. But if is necessary we can put one sentence that microscopic fungi are contaminated bread and essential oils can extend shell life of bread. The manuscript has shown how we can essential oil used in food industry.
Again authors didn’t not present here the units “MID50 and MID90 of the coriander essential oil for Penicillium expansum on the bread after 14 days 365 were 367.19 and 445.92, respectively “.
The units were added to text.

Reviewer 2 Report
General comments
This article presents the Antioxidant, antimicrobial and antibiofilm activity of coriander essential oil for its possible application in foods. The objectives of this study are clear and the data presented are interesting. Unfortunately, the manuscript gives the reader the sense of a routine study. The authors should proceed with the following revisions.
Abstract
-
The abstract has to be rewritten. It feels like a Materials and Methods section rather than an abstract. The authors should present in the Abstract the major findings of this study in order to make the article attractive and induce reader's interest to read the full text.
Introduction
-
The introduction section is balanced and needs no changes.
Materials and Methods
-
The Materials and Methods section is well-written and concise. The authors should also add how many times they have repeated each particular experiment (single, duplicate, triplicate etc) and what type of statistical analysis they have used in order to interpret their data.
Results and Discussion
General comments:
The way that the Results and Discussion section is written provides the sense that the article is relatively "poor" in results (excluding the MALDI results). Here are some suggestions for improvement:
-
I strongly suggest that the authors should separate the “Results” and “Discussion” sections.
-
The results should be analyzed more extensively. In most cases a simple number is presented!
-
The discussion in each separate section (comparison with literature) gives the sense that it is not connected to the results. Authors should compare their findings with relevant data in the literature for agreements and differences. This can be analyzed in the Discussion section.
Specific comments:
-
Table 1: How many times did the authors repeat the experiment? Is there any statistical analysis and are there any standard deviations on the relative % area, or these results are based on a single experiment? In addition, the table caption should change to "Relative chemical composition of CEO obtained by GC-MS analysis".
-
Table 1: The total amount is expected to be approximately 100%, so it has to be removed from the table.
-
Lines 240 & 243: Results and literature data should be converted in the same units in order to provide the reader instant comparison.
-
Line 252-253: What are the units? Are they cm, mm? Units should be provided.
-
Line 256: Please change “showed” to “exhibited”
-
Line 258: Please change “was found” to “has been previously reported”
-
Line 259: Please rephrase “the authors” to “the authors of that particular study”
-
Lines 265-266: MBIC refers to concentration. Though authors present results to μL/mL. These results must be converted to concentration units. If not possible to convert them to μM or a similar unit, an easy way would be to convert them to ppm or μg/mL.
-
Line 273: At which concentration did the essential oil induce inhibition in the study of Bazargani and Rohloff? How is this compared to your data?
-
Line 279: Change “in the Figure 1” to “in Figure 1”.
-
Line 279: "their" in this sentence determines spectra. Please rephrase for clarity. Figures 1 & 2: either resize the m/z values in each spectrum or create a table with all m/z values and relative intensities. Values are not obvious to the readers in the Figures presented.
-
Line 307: Change “from” to “between”
-
Line 334: “have reported”
-
Line 335: Please rephrase “Di Bonaventura et al. [33] found that the MALFI-TOF MS allows to detect” to “Di Bonaventura et al. [33] showed that the MALFI-TOF MS is a useful tool to detect”.
-
Lines 338-347: How is this extended literature review connected to your data in the present study? Please be more specific. Do your data agree and extend previous findings? Is there any difference? Please provide a comparison.
-
Line 354: Please rephrase “because of being” to “due to the fact that is”.
-
Line 366: What are the units? Units should be provided with the corresponding numbers.
Author Response
Dear reviewers and editor
RE: “Antioxidant, antimicrobial and antibiofilm activity of coriander (Coriandrum sativum L.) essential oil for its application in foods"
Manuscript ID: Foods 721388
We would like to thank the reviewers for their valuable comments and recommendations. The manuscript has been corrected in line with the comments of reviewers. All corrections are highlighted. English was improved throughout the manuscript.
Reviewer 2:
The abstract has to be rewritten. It feels like a Materials and Methods section rather than an abstract. The authors should present in the Abstract the major findings of this study in order to make the article attractive and induce reader's interest to read the full text.
Abstract was rewritten.
Introduction
The introduction section is balanced and needs no changes.
Materials and Methods
The Materials and Methods section is well-written and concise. The authors should also add how many times they have repeated each particular experiment (single, duplicate, triplicate etc) and what type of statistical analysis they have used in order to interpret their data.
The experiment was performed in triplicate, the chemical composition of the CEO was performed in duplicate and sentences for antioxidant and antimicrobial activity were added to the text of manuscript. The statistical analysis of the chemical composition of the CEO can not be performed as only one sample was used.
Results and Discussion
General comments:
The way that the Results and Discussion section is written provides the sense that the article is relatively "poor" in results (excluding the MALDI results). Here are some suggestions for improvement:
I strongly suggest that the authors should separate the “Results” and “Discussion” sections.
Chapter results and discussion was separated.
The results should be analyzed more extensively. In most cases a simple number is presented!
Results were analysed more extensively.
The discussion in each separate section (comparison with literature) gives the sense that it is not connected to the results. Authors should compare their findings with relevant data in the literature for agreements and differences. This can be analyzed in the Discussion section.
We are compare our results with literature in discussion section. Some data of coriander essential oil was first time studied.
Specific comments:
Table 1: How many times did the authors repeat the experiment? Is there any statistical analysis and are there any standard deviations on the relative % area, or these results are based on a single experiment? In addition, the table caption should change to "Relative chemical composition of CEO obtained by GC-MS analysis".
The information was added to text.
Table 1: The total amount is expected to be approximately 100%, so it has to be removed from the table.
Total amount was 99.98% it is approximately 100%. Total amount was removed.
Lines 240 & 243: Results and literature data should be converted in the same units in order to provide the reader instant comparison.
We compared and discussed our results with other authors who analyzed antioxidant activity of coriander essential oils. In this part we pointed out that other authors also reported that this oil has significant activity and is therefore also used as a natural preservative in food technology. From our opinion, converted units of these authors can be misleading, each authors had own modified method, with own calibration formula / curve and this convertion can cause problems and mistakes from this point of view. The aim of our discussion was done, we described our results and compared with other findings by free way.
Line 252-253: What are the units? Are they cm, mm? Units should be provided.
Units were provided.
Line 256: Please change “showed” to “exhibited”
It was changed.
Line 258: Please change “was found” to “has been previously reported”
It was changed.
Line 259: Please rephrase “the authors” to “the authors of that particular study”
It was changed.
Lines 265-266: MBIC refers to concentration. Though authors present results to μL/mL. These results must be converted to concentration units. If not possible to convert them to μM or a similar unit, an easy way would be to convert them to ppm or μg/mL.
The essential oil is in liquid form so we were evaluated study as μL/mL similar as different study.
Line 273: At which concentration did the essential oil induce inhibition in the study of Bazargani and Rohloff? How is this compared to your data?
MIC value in this study ranged from 0.8 to 0.63 μL/mL in different essential oils.
Line 279: Change “in the Figure 1” to “in Figure 1”.
It was changed.
Line 279: "their" in this sentence determines spectra. Please rephrase for clarity. Figures 1 & 2: either resize the m/z values in each spectrum or create a table with all m/z values and relative intensities. Values are not obvious to the readers in the Figures presented.
It was changed. Figures are bigger.
Line 307: Change “from” to “between”
It was changed.
Line 334: “have reported”
It was changed.
Line 335: Please rephrase “Di Bonaventura et al. [33] found that the MALFI-TOF MS allows to detect” to “Di Bonaventura et al. [33] showed that the MALFI-TOF MS is a useful tool to detect”.
It was changed.
Lines 338-347: How is this extended literature review connected to your data in the present study? Please be more specific. Do your data agree and extend previous findings? Is there any difference? Please provide a comparison.
The data were added.
Line 354: Please rephrase “because of being” to “due to the fact that is”.
It was changed.
Line 366: What are the units? Units should be provided with the corresponding numbers.
Units were provided.

Round 2
Reviewer 1 Report
The manuscript entitled "Antioxidant, antimicrobial and antibiofilm activity of coriander (Coriandrum sativum L.) essential oil for its application in foods" by Miroslava et al., has improved sufficiently after careful revision and hence, I recommend the manuscript for acceptance for publication.
Reviewer 2 Report
The revised manuscript merits publication in Foods.